# Uncertainty-Aware Convolutional Neural Network for Identifying Bilateral Opacities on Chest X-rays: A Tool to Aid Diagnosis of Acute Respiratory Distress Syndrome

**DOI:** 10.3390/bioengineering10080946

**Published:** 2023-08-08

**Authors:** Mehak Arora, Carolyn M. Davis, Niraj R. Gowda, Dennis G. Foster, Angana Mondal, Craig M. Coopersmith, Rishikesan Kamaleswaran

**Affiliations:** 1Department of Electrical and Computer Engineering, Georgia Institute of Technology, Atlanta, GA 30332, USA; 2Department of Biomedical Informatics, Emory University School of Medicine, Atlanta, GA 30332, USA; amonda3@emory.edu; 3Department of Surgery, Emory University School of Medicine, Atlanta, GA 30332, USA; carolyndavis@emory.edu (C.M.D.); dennis.gene.foster.iii@emory.edu (D.G.F.); cmcoop3@emory.edu (C.M.C.); 4Emory Critical Care Center, Emory University School of Medicine, Atlanta, GA 30332, USA; 5Division of Pulmonary, Critical Care, Allergy and Sleep Medicine, Emory University School of Medicine, Atlanta, GA 30332, USA; niraj.raju.gowda@emory.edu

**Keywords:** chest X-ray classification, computer-aided diagnosis, medical image analysis, model calibration, acute respiratory distress syndrome, deep learning, convolutional neural networks, uncertainty modeling

## Abstract

Acute Respiratory Distress Syndrome (ARDS) is a severe lung injury with high mortality, primarily characterized by bilateral pulmonary opacities on chest radiographs and hypoxemia. In this work, we trained a convolutional neural network (CNN) model that can reliably identify bilateral opacities on routine chest X-ray images of critically ill patients. We propose this model as a tool to generate predictive alerts for possible ARDS cases, enabling early diagnosis. Our team created a unique dataset of 7800 single-view chest-X-ray images labeled for the presence of bilateral or unilateral pulmonary opacities, or ‘equivocal’ images, by three blinded clinicians. We used a novel training technique that enables the CNN to explicitly predict the ‘equivocal’ class using an uncertainty-aware label smoothing loss. We achieved an Area under the Receiver Operating Characteristic Curve (AUROC) of 0.82 (95% CI: 0.80, 0.85), a precision of 0.75 (95% CI: 0.73, 0.78), and a sensitivity of 0.76 (95% CI: 0.73, 0.78) on the internal test set while achieving an (AUROC) of 0.84 (95% CI: 0.81, 0.86), a precision of 0.73 (95% CI: 0.63, 0.69), and a sensitivity of 0.73 (95% CI: 0.70, 0.75) on an external validation set. Further, our results show that this approach improves the model calibration and diagnostic odds ratio of the hypothesized alert tool, making it ideal for clinical decision support systems.

## 1. Introduction

Acute respiratory distress syndrome (ARDS) is a diffuse lung injury characterized by inflammation leading to increased pulmonary vascular permeability and loss of aerated lung tissue [1]. Clinical hallmarks include hypoxemia and bilateral radiographic opacities [2]. The 2012 Berlin definition of ARDS has been key in supporting clinicians and guiding clinical research [2]. However, morbidity, mortality, and healthcare costs remain unacceptably high, and ARDS remains a diagnostic challenge [3]. Studies show that early recognition of ARDS can help reduce the progression, severity, and potentially lethal clinical sequela [4].

Machine Learning (ML) methods can reliably and robustly learn complex relationships between clinical data, and several research efforts towards using ML for the predictive modeling of medical diseases are underway [5,6,7,8]. Similar research efforts for predicting the early onset of ARDS are ongoing to improve clinical recognition of the syndrome [9,10,11,12]. Although there is no single diagnostic test that can rule in or rule out ARDS, one significant limitation both clinically and with ML models is with reading chest radiographs. Interpretation of chest radiographs per the Berlin criteria can be unreliable and subject to inter-rater variability, often leading to missed or delayed ARDS diagnosis [13]. Therefore, we propose a robust and reliable ML model that can be trained to identify bilateral pulmonary opacities on chest X-ray (CXR) images, which could be an invaluable inclusion in the clinical workflow [9]. An automatic system that can raise alerts on detecting lung airspace opacities in chest X-rays of critically ill patients can identify possible ARDS cases, which a physician can then review. It can reduce the burden of reviewing CXR images in situations where there is a high patient-to-clinician ratio and can prevent cases of missed diagnoses that could occur due to human error [14,15,16].

An important requirement for using data-driven techniques like machine learning to build models for ARDS is a sizeable set of retrospective patient data with known ARDS diagnosis and onset times. Hospital billing codes are insufficient to create such a dataset due to clinical under-recognition [14]. Automated methods to filter patient encounters using constraints on clinical features from the Electronic Medical Record (EMR) like PaO_2_/FiO_2_ ratio, or positive end-expiratory pressure (PEEP) defined by the Berlin criteria are being approached [17], but are limited by non-standard documentation across EMR [18], and the heterogeneity of clinical manifestations of ARDS [19]. Thus, clinician adjudication of retrospective data is needed, which involves meticulous inspection of patient history using the EMR. An ML model that can flag patients with bilateral pulmonary opacities on chest radiographs as likely candidates for ARDS would make the adjudication process faster with higher ARDS-positive yields. Thus, our proposed tool can also speed up retrospective EMR data adjudication for ARDS cases.

Recently, there has been a surge of interest in using computer vision techniques to identify common findings on CXR images [20,21,22]. The availability of large CXR datasets such as CheXpert [20] and MIMIC-CXR [21] has made it possible to improve the performance of deep learning models. However, these datasets have labels derived from radiology notes using Natural Language Processing (NLP) Algorithms. These labels can often be inaccurate, owing to variability in radiologists’ interpretations, the language used, and differing documentation protocols [23,24,25]. Label noise can severely impact the true performance of deep learning models [26], and for clinical decision-making tasks, this degradation in performance can come at a dire cost. In this work, we have created a unique dataset of 7800 chest X-ray images with labels generated after careful adjudication of the images for bilateral pulmonary opacities by three blinded, trained physicians.

A review of the recent literature on computer-aided chest X-ray (CXR) classification was performed for the benefit of this study. Numerous studies have adopted convolutional neural networks (CNNs) to identify abnormalities on CXR images, such as pneumonia [27,28,29,30], tuberculosis [31,32,33], and COVID-19 [34,35,36,37,38,39,40,41,42,43,44]. The release of datasets like CheXpert [20] and MIMIC-CXR [21] has also enabled the training of deep learning models on large amounts of data to identify multiple CXR findings like atelectasis, edema, consolidation, cardiomegaly, and pleural effusion. Several high-performing models utilize transfer learning by pretraining CNNs on the ImageNet dataset and fine-tuning them on the target dataset [45]. Notable architectures used for this purpose include ResNets [46,47], DenseNets [20,48,49], Swin Transformers [29], and ConvNeXts [37,50]. Irvin et al. [20] use the Densenet121 architecture to achieve an AUC of 0.90 on the multi-class classification problem. Yuan et al. [48] train models on the CheXpert dataset with the DenseNet121, DenseNet161, DenseNet169, DenseNet201 architectures using a Deep AUC Maximization loss. The DenseNet121 model achieves a superior AUC of 0.93 for a five-class chest X-ray abnormality classification model. Among other notable models, DarkCovidNet [38], based on the DarkNet-19 model used in YOLO object detection, achieves an accuracy of 0.98 for COVID-19 detection. Nahiduzzaman et al. [44] employ a light-weight convolutional neural network coupled with an extreme learning machine model to classify chest X-ray images into 14 classes with an average AUC of 0.96. Yao et al. [51] use the DenseNet121 as a deep image feature extractor and apply an LSTM (Long Short-Term Memory) network to exploit dependencies between labels, achieving an AUC of 0.79 on a 14-class classification problem using the ChestX-ray14 dataset. Islam et al. [52] use a combined CNN-LSTM framework for COVID-19 detection and achieve an accuracy of 0.99.

Many works incorporate attention mechanisms into CNNs to focus on regions of interest and improve diagnostic accuracy [32,47,49,53]. Many studies use ensembles of machine learning classifiers to improve disease classification using CXR images [29,40,43]. Zhao et al. [49] achieves an AUC of 0.85 on a 14-class chest X-ray classification task, using an attention module added to the DensNet121 architecture and the focal loss for combating class imbalance. Various image processing and augmentation techniques have been used to improve performance and generalization of deep learning models [36,54,55]. Image preprocessing steps include morphological operations for region-of-interest segmentation, machine learning-based lung segmentation, histogram equalization, low-pass filtering, and contrast enhancement [45]. An emerging field of interest involves using generative models to create artificial chest X-ray images for augmenting training datasets. Some researchers have used generative adversarial networks (GANs) for this purpose [56,57], while others have explored multi-modal diffusion models to generate chest X-ray images based on text prompts [58]. These approaches show promise in further enhancing the capabilities of deep learning models for CXR classification.

Research efforts have also been made to specifically diagnose ARDS [10,12,59]. In the work of Reamaroon et al. [10], an image processing-based feature engineering was used in conjunction with deep learning for ARDS detection. Sjolding et al. [12] used a CNN to achieve high performance in identifying ARDS from chest X-rays, and Pai et al. [59] used a multi-modal ensemble framework combining clinical data with chest X-ray imaging to predict ARDS in the first 48 h of admission. A method for dealing with ARDS label uncertainty is proposed by Reamaroon et al. [60], where labels used to train the machine learning model have a confidence score reflecting expert uncertainty in ARDS diagnosis. A similar approach was also followed in the work of Sjolding et al. [12] for ARDS identification. Owing to the complex nature of the syndrome and its diagnosis, we do not attempt to identify ARDS from CXR images in our work but instead identify bilateral pulmonary opacities from CXR images. Previous studies have also worked towards identifying specific lung opacities [23,61]. Vardhan et al. [23] quantify the extent of lung opacities in CXR images, while Kim et al. [61] propose a method to localize opacities in the four quadrants of the lung. In this work, we trained CNN to predict three classes: “bilateral opacities present”, “bilateral opacities absent”, and “equivocal”. Many previous works in CXR image classification disregarded equivocal images or treated them as controls. In contrast, we allow our model to predict the “equivocal” class. We use an uncertainty-aware label-smoothing technique that improves model calibration. Both these training methods ensure that the CNN does not make overconfident predictions on images it is unsure about and can defer these to the physician, making it easy to incorporate in the clinical workflow and potentially improve the rate of ARDS detection [9]. Thus, our model served as a proof of concept of a valuable tool for generating proactive alerts, allowing for the early identification of potential Acute Respiratory Distress Syndrome (ARDS) cases in clinical and research settings.

## 2. Materials and Methods

### 2.1. Dataset Generation

This was a single-center retrospective cohort study at an academic institution. We included de-identified patients diagnosed with sepsis based on the Sepsis 3 criteria, admitted to the Surgical and Medical Intensive Care Units (ICU) at Emory University Hospital between August 2015 to May 2019. All patients chosen were over 18 years of age. A total of 7800 single-view frontal chest radiographic images and their corresponding radiologist-dictated reports, corresponding to 664 patient encounters, were extracted from our Electronic Medical Record (EMR) database. These images were annotated by three blinded physicians with critical care experience for pulmonary opacities using the labels: “bilateral”, “left lung”, “right lung”, or “absent”. Unclear images, images with occluded lung regions, or images that could not be interpreted with certainty without further information about a patient’s clinical course, were labeled “equivocal”. Inter-rater disagreements were resolved conservatively, biased towards limiting false positives. If two annotators agreed, their label would be considered the “ground truth label”. If all three clinicians agreed that opacities were present, the image was labeled “bilateral opacities present”. If all three clinicians disagreed on the image’s label, the image was labeled “equivocal”.

### 2.2. Training the Convolutional Neural Network

#### 2.2.1. Image Preprocessing

We selected all lung-window single-view chest X-ray scans of patient encounters in the chosen cohort and converted them from DICOM format to PNG files. To ensure consistent image analysis, we normalized the image intensity histograms of all chest X-ray images. During the training process of our network, we applied various on-the-fly data augmentation techniques, including random rotations, left-right flips, affine shifts, scaling, and random brightness and contrast adjustments. These augmentation techniques were employed to simulate the inherently noisy nature of chest radiograph images of critically ill patients in the intensive care unit (ICU). Such patients may be unable to maintain a straight posture during the X-ray procedure or may have occluding objects and support devices that cannot be removed. By incorporating these transformations into the training process, we aimed to enhance the robustness of our network to handle real-world variations encountered in ICU CXR scans. During inference and testing, the images were resized to 256×256, histogram normalized, and contrast adjusted using γ=1.5.

#### 2.2.2. Network Architecture and Training Parameters

We employed a Convolutional Neural Network (CNN) with the Densenet121 architecture [62], augmented by a fully connected layer, to categorize chest X-ray images into three distinct classes: “bilateral opacities present”, “bilateral opacities absent”, and “equivocal”. We pre-trained the network on the ImageNet dataset to facilitate the training process. Subsequently, we fine-tuned the CNN on the CheXpert dataset, which comprises approximately 191,000 chest X-ray images and associated labels derived from radiology notes. We trained the CNN to identify eight common findings, i.e., lung opacities, edema, consolidation, pneumonia, atelectasis, pneumothorax, pleural effusion, and support devices, using the CheXpert data labels. We further fine-tuned the network on our cohort of 7800 ground-truth annotated images for our specific task, utilizing a patient-level train-test split of 70–30. We used the Adam optimizer [63] with a learning rate of $10^{-4}$ and trained the CNN for 30 epochs. To address the class imbalance, we utilized a weighted random sampler that combined undersampling of the majority class and upsampling of the minority class, utilizing the class-balanced weights function from the sci-kit-learn Python package. We experimented with different loss functions, including focal loss [64] with a γ=2, cross-entropy loss, cross-entropy loss with label smoothing [65], and our proposed cross-entropy loss with uncertainty-aware probability targets. The high-level training diagram of our network is shown in Figure 1. To compare our three-class prediction approach with standard training schemes of predicting only the positive and negative class, we conducted similar experiments by training the CNN to predict only two classes, namely “bilateral opacities present” and “bilateral opacities absent”, while disregarding equivocal images as well as while treating them as controls. The outcomes are presented in detail in Section 3 of this report.

We perform additional experiments using different model architectures, as well as ensembling techniques to improve the classification performance of our network [50,66,67]. Details and results about additional experiments we performed to improve diagnostic accuracy are provided in Appendix A.

#### 2.2.3. Cross-Entropy Loss with Uncertainty Aware Probability Scores

If $p_{k}=\frac{e^{x^{k}w_{k}}}{\sum_{l=1}^{K}e^{x^{T}w_{l}}}$ is the likelihood of the *k*-th class out of K classes, the cross-entropy loss minimizes the function $H\left(y,p\right)=\sum_{k=1}^{K}-y_{k}\log\left(p_{k}\right)$ where $y_{k}$ are one-hot encoded class labels, with “1” for the correct class and “0” for all other classes. For a network with label smoothing with parameter α, the smoothed cross-entropy loss minimizes the following function $H^{s}\left(y^{s},p\right)=\sum_{k=1}^{K}-{y^{s}}_{k}\log\left(p_{k}\right)$ where $y^{S}=y\left(1-\alpha \right)\;+\alpha /K$ [65]. Such soft targets are shown to improve model calibration so that the confidence of predicted probabilities is proportional to the likelihood of the prediction being accurate [65]. For our three-class model predicting bilateral opacities present, absent, and equivocal images, we tweak the soft targets to reflect the probabilities of the respective classes. In essence, if the ground truth label is “equivocal”, the target likelihoods of the “present” and “absent” classes are set to 0.5. Labels are one-hot encoded by assigning a (1, 0, 0) target label if bilateral opacities are absent, a (0, 1, 0) target label if bilateral opacities are present, and a (0.5, 0.5, 1) target label if the image is marked equivocal. Our results show that this training method allows the CNN to predict the equivocal class for uncertain input images, reducing the rate of false positives and false negatives while maintaining good sensitivity and precision. The CNN trained with uncertainty-aware probability scores performed the best and was chosen for further analysis and external validation.

### 2.3. Performance Metrics

We conducted a comparative analysis of our convolutional neural network models based on standard performance metrics. These metrics include the Area under the Receiver Operating Characteristics Curve (AUROC) and the Precision–Recall Curve (AUPRC), balanced accuracy score, precision, sensitivity, specificity, F-score, and the Diagnostic Odds Ratio (DOR). These are defined as follows:(1)$$\mathrm{precision}=\frac{\mathrm{True}\;\mathrm{Positives}}{\mathrm{True}\;\mathrm{Positives}+\mathrm{False}\;\mathrm{Positives}}$$
(2)$$\mathrm{sensitivity}=\frac{\mathrm{True}\;\mathrm{Positives}}{\mathrm{True}\;\mathrm{Positives}+\mathrm{False}\;\mathrm{Negatives}}$$
(3)$$\mathrm{specificity}=\frac{\mathrm{True}\;\mathrm{Negatives}}{\mathrm{True}\;\mathrm{Negatives}+\mathrm{False}\;\mathrm{Positives}}$$
(4)$$\mathrm{balanced}\;\mathrm{accuracy}=\frac{\mathrm{sensitivity}+\mathrm{specificity}}{2}$$
(5)$$\mathrm{DOR}=\frac{\mathrm{sensitivity}\times \mathrm{specificity}}{\left(1-\mathrm{sensitivity})\times (1-\mathrm{specificity}\right)}$$

These metrics capture a more comprehensive view of model performance and utility in the clinical workflow. The diagnostic odds ratio ranges from zero to infinity, and higher diagnostic odds ratios indicate better test performance [68]. The balanced accuracy score is used as it is a better estimate of model performance in the case of multi-class classification with class imbalance.

### 2.4. Model Calibration

In assessing the test performance of our models, we considered model calibration as a crucial criterion. Calibration refers to how well the predicted likelihoods from the model align with the actual confidence of the predictions being accurate. To quantify the calibration, we utilized confidence binning and calculated the Maximum Calibration Error (MCE). To apply confidence binning, the predicted likelihoods generated by the model are divided into ten bins at regular. For each bin, we computed the average prediction likelihood and accuracy of the predictions falling within that bin. The MCE, the maximum difference between the average predicted likelihood and the average accuracy across all bins was chosen as the calibration measure. This metric indicates the worst-case calibration error, making it particularly relevant for clinical decision-making tasks where reliable confidence measures are essential.

### 2.5. Model Interpretability

We employed two interpretability techniques for deep image classification, namely Grad-CAM [69] (Gradient-weighted Class Activation Mapping) and Occlusion Sensitivity Maps [70], to understand the classification decisions made by our CNN model. Grad-CAM is a visualization method that generates heatmaps, indicating the regions of an image most influential in the CNN’s classification decision. Gradients of the target class score with respect to the feature maps from the final convolutional layer of the CNN are calculated and used to assign weights to the feature maps, resulting in a heatmap highlighting the discriminatory regions. Occlusion Sensitivity Maps involve gradually occluding different parts of an image and observing the resulting changes in the CNN’s output. This technique helps identify the regions within the image that significantly contribute to the CNN’s classification decision and regions that decrease the CNN prediction confidence.

### 2.6. External Dataset Validation

We use the MIMIC-CXR dataset [21] for external validation of our model. To do so, we first matched chest X-ray images and radiology notes to all patients from the MIMIC-IV dataset that were diagnosed with sepsis-3. We then randomly sampled approximately 1600 images from this subset and obtained physician labels using the same strategy described in Section 2.1. These are considered ground truth labels for validating our algorithm on the external dataset. The chest X-ray scans undergo the same pre-processing steps and test time transformations as those applied to our internal testing dataset. These include resizing the image to a 256×256, histogram normalization, and contrast adjustment to a γ=1.5.

### 2.7. Comparison with Labels Derived from Radiology Notes

Our study used a rule-based Natural Language Processing (NLP) pipeline to analyze the radiology notes associated with each chest X-ray scan. We followed the methodology outlined by Irvin et al. [20] and designed an NLP pipeline using the medspaCy [71] library, a framework specifically tailored for clinical text processing. The medspaCy pipeline incorporated various components to facilitate the analysis. Firstly, we employed the in-built sentencizer module for text preprocessing, enabling the segmentation of the radiology notes into individual sentences. We then performed Named Entity Recognition using a target matcher component to identify specific entities of interest within the text. Additionally, we integrated a context module into the pipeline to capture the contextual information related to the identified named entities, such as negation or likelihood. To structure the analysis, we included a sectionizer component as a pre-processing step to demarcate different sections within the radiology notes, including clinical indications, impressions, and findings pertaining to the lungs/pleura, heart/mediastinum, and bones/soft tissue. Additional context and target rules derived from the language syntax frequently seen in our cohort of radiology notes were incorporated, as were phrases and context rules described by the Stanford CheXpert Labeler team. For the identification of possible cases of pulmonary opacification, we employed specific keywords, including ARDS, atelectasis, consolidation, edema, effusion, opacity, infiltrates, pneumonia, and pneumothorax. These keywords were used to flag instances in the radiology notes that potentially indicated the presence of lung opacities, and the associated images were subsequently filtered out based on mentions of these findings in associated radiology notes. They were then categorized into positive (indicating the presence of the finding), negative (indicating the absence of the finding), or uncertain (suggesting the possibility of existence, requiring further clinical adjudication). Furthermore, we detected disease descriptors, such as ‘bilateral’ for ‘opacity’, ‘infiltrates’, and ‘pleural’ or ‘interstitial’ for ‘effusions’ or ‘edema’, respectively. To label bilateral opacities, we utilized descriptors of laterality (’Bilateral’, ‘Right’, ‘Left’). Instances describing findings as only ‘Right’ or ‘Left’ were labeled negative for bilateral opacities. We compared the performance of the labels derived from these radiology notes with the output of our CNN model, which was trained on physician labels. The obtained results are discussed in detail in Section 4 of our study.

## 3. Results

### 3.1. Data Characterization

The number of patients, chest radiographs, image label frequency, and demographic information of the internal and external datasets are listed in Table 1. Class labels in both datasets followed a long-tailed distribution, with the lowest prevalence of the equivocal class. Since our patient cohort were all those diagnosed with sepsis, we can see a higher incidence of the bilateral pulmonary opacities present class when compared to the absent and equivocal classes, with 54% present in the internal (Emory University) dataset and 77% present in the external (MIMIC-CXR) dataset.

### 3.2. CNN Performance Comparison

Table 2 reports the performance metrics of the evaluated training schemes. All the CNN models compared are those pre-trained on the ChexPert dataset and then had all layers fine-tuned on the Emory Dataset. These models saw a superior performance to those trained from scratch, those pre-trained on the CheXpert dataset, and those with their last layers fine-tuned on the Emory dataset. The metrics in Table 2 and the Receiver Operating Characteristics (ROC) curves and Precision–Recall Curves (PRC) in Figure 2 are calculated for the “Bilateral Opacities Present” class to enable comparison between the two-class and the three-class approach. The three-class model refers to the training scheme in which we train the model to predict the “present” or “absent” as well as the “equivocal” class. The two-class model refers to the training scheme in which we predict only the “present” and “absent” classes. Images labeled “equivocal” as belonging to the “absent” class in one set of experiments were disregarded and removed from analysis for a second set.

The results in Table 2 show that the highest performance metrics were observed when testing the two-class model (disregarding equivocal). However, this might represent a biased testing set as we eliminated the difficult examples from the test set. Thus, this approach was not considered for performance comparison or external validation. The three-class model outperforms the two-class model (with equivocal images treated as controls) in precision, AUPRC, and diagnostic odds ratio. Superior performance was achieved by training the three-class model using the cross-entropy loss with uncertainty-aware probability targets, with an AUROC of 0.828 (95% CI: 0.803, 0.853) and F-score of 0.746 (95% CI: 0.720, 0.775), and a diagnostic odds ratio of 17.211 (95% CI: 12.106, 25.976). The Receiver Operating Characteristics curves (ROC) and Precision–Recall Curves (PRC) in Figure 2 highlight the benefit of our three-class approach in improving the precision and recall of the trained model. The benchmark performance of labels derived from radiology notes showed an average precision of 0.63, an average sensitivity of 0.58, and an average specificity of 0.81. Results of a 10-fold cross-validation are provided in Appendix A.

### 3.3. Model Calibration

Table 3 reports the maximum calibration error for the experiments described in the previous section. Figure 3 plots reliability diagrams that depict model calibration. It is observed that the three-class model trained using the cross-entropy loss with probability targets has the lowest MCE of 0.150 with the best-calibrated reliability diagram. The three-class model trained with focal loss is a close second, with an MCE of 0.240. It can be noticed that the two-class models have inferior model calibration when compatible with the three-class models.

### 3.4. External Validation on MIMIC-CXR

The three-class model trained with the cross-entropy loss using uncertainty-aware probability targets was chosen as the best-performing model and used for external validation. Table 4 reports the performance metrics of the model on the external validation set, while Figure 4 plots the Receiver Operating Characteristics curves and Precision–Recall Curves for the same. The model achieves an AUROC of 0.836 (95% CI: 0.811, 0.858), an F-score of 0.658 (95% CI: 0.627, 0.685), and a diagnostic odds ratio of 2.18 (95% CI: 1, 1.8). Table 1 shows us that the external validation set used for our study has a much higher incidence of the “Bilateral Opacities Present” class. This could explain the reason behind observing a higher AUC and specificity but a lower sensitivity and diagnostic odds ratio. The benchmark performance of labels derived from radiology notes showed an average precision of 0.66, an average sensitivity of 0.59, and an average specificity of 0.58.

### 3.5. Visualization of Saliency Maps

By utilizing GradCAM and occlusion sensitivity maps, we aimed to gain insights into the decision-making process of our CNN model and understand the specific regions of the input images that were most influential in determining the classifications. These visualization techniques enhance our understanding of the model’s reasoning and aid in interpreting its predictions. In Figure 5, we can see that while the Grad-CAM either focuses on the entire lung region or each lung separately, the occlusion sensitivity maps show us the exact regions of the image that were important when making the positive prediction. These regions correspond well with opacified lung spaces. In Figure 5a, an interesting observation is that the Grad-CAM clearly pays attention to both lungs and correctly classifies the image as negative for bilateral opacities, despite one lung looking opacified. In image sets (c) and (f), the grad-CAM activations focus on lung regions that are occluded or difficult to interpret with certainty due to poor quality. Figure 6 displays instances of misclassification compared to the gold standard of physician labels. Image sets (c) and (d) are examples of the CNN classifying images that are labeled equivocal as “Bilateral Opacities Absent” and “Bilateral Opacities Present”, respectively, albeit with lower likelihoods. Image sets (e) and (f) are examples of the CNN classifying certain images that it is uncertain about as “equivocal”. We notice that the Grad-CAM appears to be focusing on the breast tissue region, which might confuse the predictions of airspace opacities.

## 4. Discussion

Recent research on ARDS has moved towards early recognition as a means of reducing severity and mortality, and the initiation of therapies known to reduce the likelihood of ARDS development [72], like low tidal volume ventilation [73] and fluid restriction [74]. Critically ill patients have frequent chest X-rays taken to ensure the correct placement of devices such as chest tubes, central lines, or pacemakers, and for routine medical workups [75]. Using these images to help with the accurate and early identification of bilateral pulmonary opacities can optimize patient care and ongoing clinical management, possibly improving the chances of detecting ARDS in the early stages [9]. In this work, we have proposed a CNN trained to identify pulmonary opacities on chest X-ray images as a tool to generate predictive alerts for possible ARDS cases that can be applied in the clinical and research domains. The overarching goal of our research was to develop a robust and dependable model capable of detecting bilateral opacities on chest radiographs to translate it into clinical settings. The clinical utility of our work is threefold.

First, we have collected a dataset comprising all chest X-ray (CXR) images of patients diagnosed with sepsis-3 during their hospitalization at Emory University Hospital. We did not curate the dataset by selectively choosing good-quality images or by sampling images from specific ‘present’ and ‘absent’ classes. This approach stands in contrast to some recent studies focusing on disease detection, in which datasets are composed of positive disease samples, and controls [34,35,38,52]. Some studies struggle with poor generalization when applied in real clinical settings [76,77]. The lack of representation of critical patients with possible differential diagnoses and the differences in inclusion criteria for datasets can be confounding factors that contribute to this disparity in performance. Instead, we aimed to maintain a reflection of real-world data distributions by including all available images. Consequently, our patient cohort represents a high-acuity group, and a significant portion of the images received equivocal labels as determined by the clinical adjudicators (23%). Images in our data are often of poor quality, as they are often captured at the bedside in the ICU, with support devices, chest tubes, and other occluding objects.

Second, we generate ground truth labels for all images from three blinded clinicians to train the network instead of using labels extracted from radiology notes as these might not always be accurate or reflective of a patient’s current clinical status [15,24,25]. Findings that are repetitive or beyond the indication, such as line placement confirmation, often go unlabeled completely. In the case of labeling for bilateral opacities, the problem becomes more challenging owing to variability in the interpretation of the criterion. Additionally, radiologists may omit mentions of laterality or might use non-standard language to indicate findings. We have also observed similar evidence of unreliable labels from radiology notes, which can be seen in Figure 7. In this work, we create a first-of-its-kind dataset of single-view chest X-ray images of critically ill patients annotated for bilateral opacities, unilateral opacities, or labeled equivocal by three clinical experts accounting for inter-rater variability. Since the presence of bilateral opacities on chest radiographs is one of the primary criteria of the Berlin Definition for ARDS, such a dataset could be pivotal in predictive modeling for ARDS using machine learning algorithms. The rigorous process of clinician adjudication guarantees that the performance metrics reported in our study hold clinical significance and can be trusted in real-world medical applications. By relying on ground truth labels established through consensus among multiple experts, we provide a reliable foundation for training our network and validating its performance.

Third, our machine learning algorithm attempts to incorporate trust and uncertainty awareness by modeling the “equivocal” class. We allow clinicians to label images as equivocal and design a three-class training scheme incorporating label uncertainty and explicitly teaching CNN to identify the equivocal class. This approach was adopted to develop a reliable model that can label an image as ‘equivocal’ rather than producing low-confidence predictions on uncertain cases. By training our model to handle equivocal labels, we aimed to create a system that can be seamlessly integrated into the clinical workflow or employed in a human-in-the-loop setting. Unlike many current state-of-the-art studies, which focus on improving the AUROC, our training goal was to produce a well-calibrated model, incorporating uncertainty awareness via the “Equivocal” class. The output probabilities of a well-calibrated model are good estimates of its predictive confidence. This ensures that the model’s predictions are reliable and can be interpreted with greater confidence in real-world clinical applications. To demonstrate the superiority of our three-class model over a two-class model that treats equivocal labels as “bilateral opacities absent”, we conducted an experimental comparison. The three-class model showed higher AUROC, Area Under the Precision–Recall Curve (AUPRC), F-score, calibration error, and diagnostic odds ratio compared to the two-class model. This indicates that our approach of explicitly handling equivocal cases leads to better performance in various evaluation metrics. Moreover, we evaluated different loss functions and found that our uncertainty-aware cross-entropy loss with probability targets achieved a lower maximum calibration error when compared to focal loss and standard cross-entropy loss. This is evident from our results in Table 3 and Figure 3, further validating the effectiveness of our training scheme. The diagnostic odds ratio balances a trade-off between sensitivity and specificity, thereby balancing the trade-off between false positive rates (probability of false alarms) and false negative rates (miss rates). This measure helps evaluate the diagnostic accuracy of the models. Our results show that the three-class model has higher precision, sensitivity, AUPRC, and diagnostic odds ratio when compared to the corresponding two-class models (Table 2).

On-par classification performance on the external validation dataset reiterated the robustness of our approach. We noticed an increase in validation performance in terms of the AUC, which could be an artifact of the much higher number of positive class cases in the external test set. Grad-CAM and occlusion sensitivity map visualizations were used to lend interpretability to class predictions. Figure 5 and Figure 6 show us that the CNN is picking up on the regions of the lungs that are important in identifying bilateral opacities. This confirms that the CNN is trained to look at the relevant areas of the image instead of finding “shortcuts” by picking up accidental statistical correlations with irrelevant regions in the images, which is a growing concern with large datasets from a limited number of hospital centers [77]. Similar saliency maps for the external dataset prove the reliability and robustness of our model.

Our study has some limitations that need to be addressed. First, we limited our dataset to chest X-ray images from a cohort of patients diagnosed with sepsis. This was done to prevent the inclusion of trauma cases, and other causes of ARDS, whose chest X-rays might look widely varied. In future work, we plan to include all critically ill patients admitted to the ICU to train our models on a more diverse patient population. Second, we obtain gold-standard ground truth labels on the external MIMIC-CXR dataset in the same method as the internal validation set due to the lack of pre-available ground truth labels for “bilateral” chest X-ray opacities. Such an external validation might not account for inherent bias in our labeling scheme. However, since three blinded clinicians provide our labels, we consider this an unbiased physician-annotated X-ray read. Another limitation of work might be that our dataset is annotated by physicians instead of radiologists. However, all three physicians have critical care experience. In clinical practice, images obtained at the bedside are often interpreted in real time and factor in the patient’s current clinical status, which is often not reflected in the electronic medical record in real-time. Additionally, collaborative discussions between critical care providers and radiologists are not reflected in radiographic annotations. Taken together, the advantage of training our model using annotated images by three critical care-trained physicians is more reflective of actual clinical practice, potentially improving the translatability of our model into clinical practice.

Moreover, this work is a proof-of-principle of an automated tool that can be used to check the chest radiograph criteria of the Berlin definition of ARDS. However, ARDS is a syndrome that needs a multi-modal approach for diagnosis. As a next step, we aim to look at a series of chest X-ray scans of a particular patient and integrate clinical vitals, labs, and blood gas readings to build a better predictive model for ARDS diagnosis.

In future work, the inclusion criteria for our study will be explained to include CXR images of all patients admitted to the Emory University Hospital from 2016–2022, to enhance accuracy in our model. We will explore using generative models to augment the training data by producing artificially generated CXR image samples. Additional considerations to this approach include acquiring clinician-validated “ground-truth” labels for artificially generated images. Attention-based convolutional neural networks will be considered, in addition to transformer-based methods to improve performance.

## 5. Conclusions

Acute Respiratory Distress Syndrome (ARDS) is a serious lung injury with high mortality that is often missed or diagnosed late, leading to poor patient outcomes. An important criterion for diagnosing ARDS is the presence of bilateral pulmonary opacification. In this work, we created a unique dataset of single-view CXR images of critically ill patients labeled as bilateral opacities, unilateral opacities, and equivocal by three clinical experts with inter-rater agreement. Our model is robust to clinically equivocal images and explicitly flags images as equivocal, indicating the need for more information or physician input. Our training approach using uncertainty-aware soft likelihood targets provides reliable output probabilities calibrated to confidence in the presence of bilateral pulmonary opacities. The CNN model identifies bilateral opacities with high precision and a low false-positive and false-negative rate. Thus, our model design is well-suited for an automated, always-on alert system that can tirelessly read and make predictions on chest X-rays with high confidence while deferring equivocal images to clinicians on-call. Such an AI-clinician collaborative strategy for ARDS onset prediction can be a useful tool for early identification and for retrospective adjudication for data-driven studies.

## Figures and Tables

**Figure 1 bioengineering-10-00946-f001:**
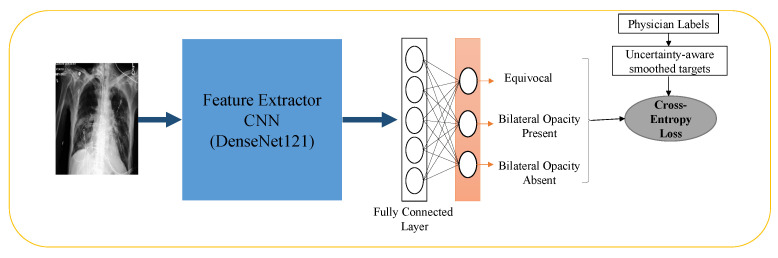
High-level training diagram of the convolution neural network.

**Figure 2 bioengineering-10-00946-f002:**
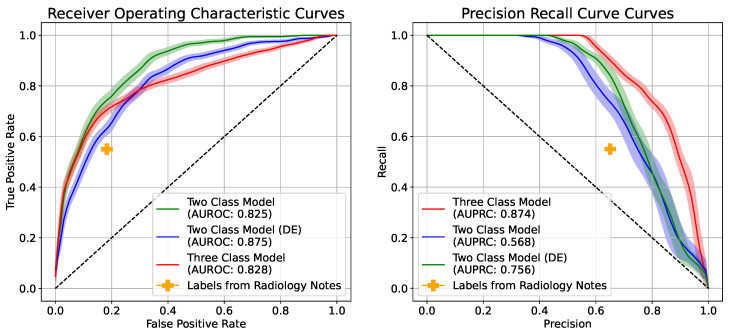
Receiver Operating Characteristics curves (ROC) and Precision–Recall Curves (PRC) for the positive class (bilateral opacities present) of the two-class and three-class models, as well as the benchmarked performance of the labels derived from radiology notes using an NLP algorithm. DE: Disregarded Equivocal images.

**Figure 3 bioengineering-10-00946-f003:**
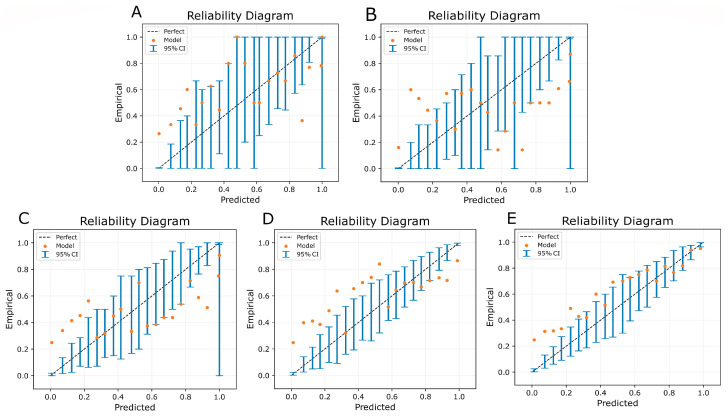
Reliability diagrams to inspect model calibration for (**A**) two-class model trained with cross-entropy loss, (**B**) two-class model trained with focal loss, (**C**) three-class model trained with cross-entropy loss, (**D**) three-class model trained with focal Loss, (**E**) three-class model trained with cross-entropy loss using uncertainty-aware probability targets.

**Figure 4 bioengineering-10-00946-f004:**
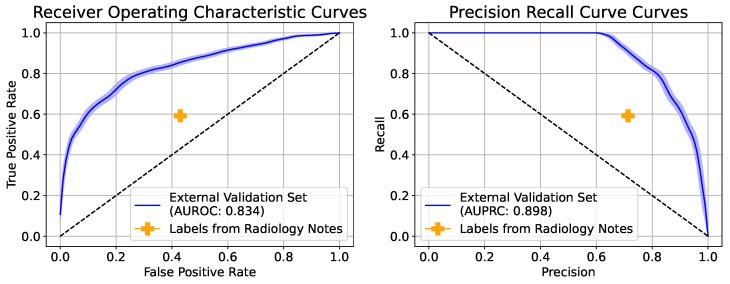
Receiver Operating Characteristics curves (ROC) and Precision–Recall Curves (PRC) for the positive class (bilateral opacities present) of three-class Model trained with cross-entropy loss using uncertainty-aware probability targets on the MIMIC-CXR external validation set, as well as the benchmarked performance of the labels derived from radiology notes using an NLP algorithm.

**Figure 5 bioengineering-10-00946-f005:**
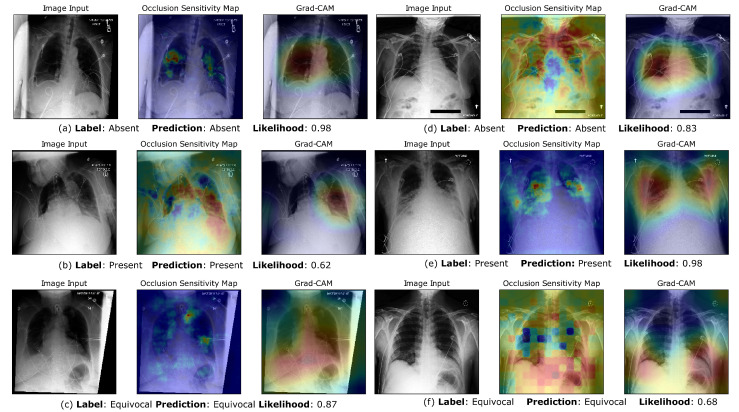
Occlusion Sensitivity Maps and Grad-CAM visualizations of network predictions for correctly classified examples. (**a**–**c**) are CXR images belonging to the internal test set, while (**d**–**f**) are CXR images belonging to the external validation set. For the Grad-CAM visualizations, the red highlighted regions have the highest discriminatory importance for the predicted class. For the Occlusion Sensitivity Maps, the hotter (red) regions are the areas that increased model confidence in the predicted class. In contrast, the cooler (blue) regions were the most confusing to the CNN (increased predicted class likelihood when occluded).

**Figure 6 bioengineering-10-00946-f006:**
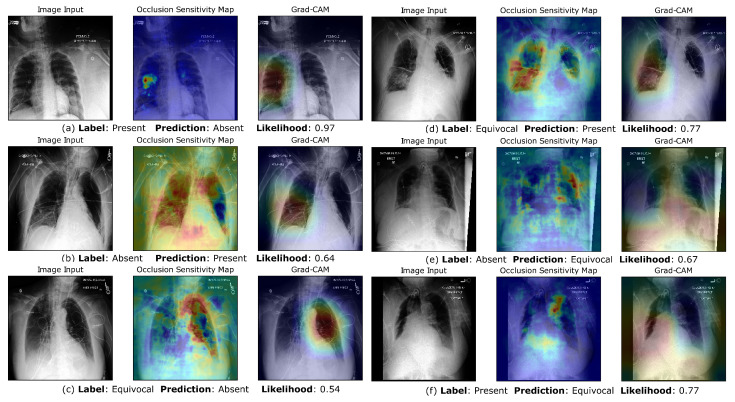
Occlusion Sensitivity Maps and Grad-CAM visualizations of network predictions for misclassified examples. (**a**,**b**) Examples of misclassification between the “Bilateral Opacities Present” and “Bilateral Opacities Absent” Class. (**c**,**d**) Examples of CXR images labeled “equivocal” by clinicians, that the network identifies as “Bilateral Opacities Present” and “Bilateral Opacities Absent”. (**e**,**f**) Examples of CXR images labeled “Bilateral Opacities Present” and “Bilateral Opacities Absent”, that the network identifies as “equivocal”.

**Figure 7 bioengineering-10-00946-f007:**
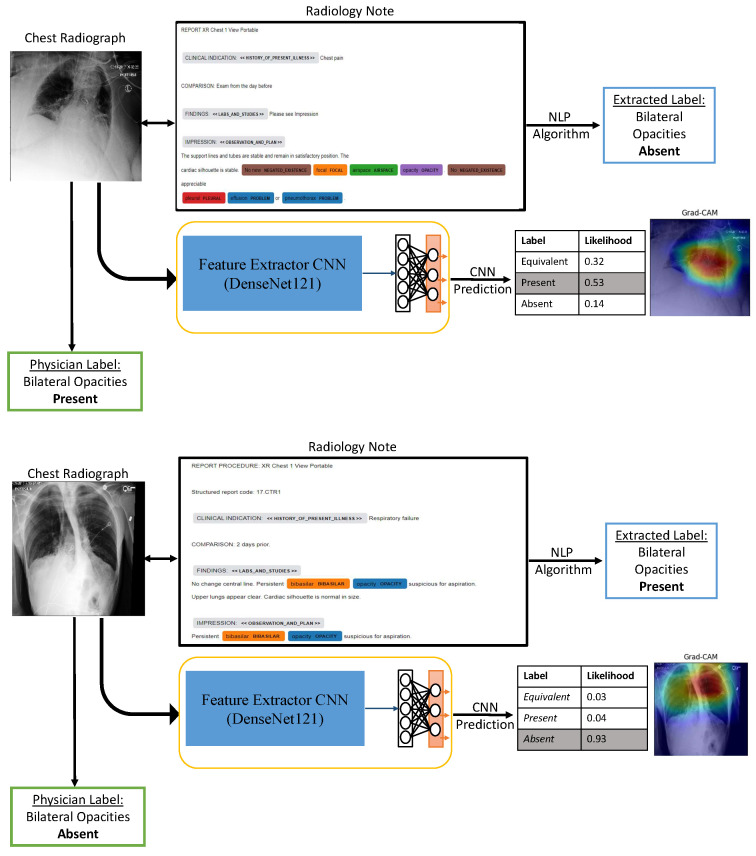
Illustrative examples of incorrect labels derived from radiology notes.

**Table 1 bioengineering-10-00946-t001:** Demographic information of the internal test set from Emory University and the external validation set (MIMIC-CXR). Data are reported as the number of counts for patients and CXR images, number of counts (percentage of total counts) for sex, race, and label prevalence, or median (IQR) for Age.

		Internal Dataset (Emory University)	External Dataset (MIMIC-CXR)
**Patients**		663	952
**Chest X-rays**		7825	1639
**Age—median (IQR)**		57 (43–67)	65 (53–76)
**Sex**	Male	315 (48%)	532 (56%)
	Female	348 (52%)	420 (44%)
**Race**	Caucasian/White	304 (46%)	605 (63%)
	African American/Black	251 (38%)	153 (16%)
	Asian	14 (2%)	44 (5%)
	Other	94 (14%)	150 (16%)
**CXR Labels**	Bilateral Opacities Present	4227 (54%)	1009 (61.5%)
	Bilateral Opacities Absent	1788 (23%)	442 (27%)
	Equivocal	1810 (23%)	188 (11.5%)

**Table 2 bioengineering-10-00946-t002:** Performance metrics of the two-class model (predicting “present” or “absent”) with equivocal images considered “absent”, two-class model (disregarding equivocal) with equivocal images disregarded from training and testing, and the three-class model (predicting “present”, “absent” and “equivocal”). The highest values for each metric are emboldened, excluding the two-class model (disregarding equivocal) to ensure valid and unbiased comparisons. A 95% CI is reported and was calculated by bootstrap resampling 1000 times.

Experiment	Loss Function	AUROC	AUPRC	F-Score	Precision	Sensitivity	Specificity	Diagnostic Odds Ratio	Balanced Accuracy
Two-Class Model	Cross-Entropy Loss	0.819 (0.791, 0.844)	0.553 (0.521, 0.584)	0.707 (0.679, 0.735	0.678 (0.671, 0.726)	0.724 (0.694, 0.754)	0.809 (0.781, 0.836)	11.141 (8.545, 14.765)	0.767 (0.736, 0.795)
	Focal Loss	0.825 (0.802, 0.852)	0.568 (0.539, 0.595)	0.715 (0.684, 0.744)	0.706 (0.674, 0.735)	0.730 (0.698, 0.732)	0.816 (0.787, 0.843)	11.967 (8.979, 16.144)	0.773 (0.743, 0.788)
Two-Class Model^*DE*^	Cross-Entropy Loss	0.884 (0.864, 0.904)	0.758 (0.736, 0.779)	0.734 (0.689, 0.743)	0.777 (0.750, 0.807)	0.714 (0.689, 0.743)	0.923 (0.904, 0.941)	29.917 (22.455, 43.151)	0.819 (0.796, 0.842)
	Focal Loss	0.875 (0.855, 0.896)	0.756 (0.731, 0.789)	0.774 (0.746, 0.799)	0.775 (0.750, 0.803)	0.772 (0.742, 0.801)	0.868 (0.843, 0.893)	22.235 (16.398, 31.375)	0.820 (0.793, 0.847)
Three-Class Model	Focal Loss	0.810 (0.785, 0.837)	0.845 (0.803, 0.8475)	0.739 (0.712, 0.766)	0.738 (0.711, 0.766)	0.740 (0.712, 0.767)	0.712 (0.666, 0.757)	7.006 (5.011, 10.191)	0.726 (0.689, 0.762)
	Cross-Entropy Loss	0.790 (0.764, 0.818)	0.819 (0.0.792, 0.834)	0.688 (0.659, 0.717)	0.716 (0.691, 0.744)	0.707 (0.681, 0.732)	**0.851 (0.819, 0.883)**	13.769 (9.948, 19.926)	0.776 (0.750, 0.808)
	Cross-Entropy Loss with probability targets	**0.828 (0.803, 0.853)**	**0.874 (0.861, 0.887)**	**0.746 (0.720, 0.775)**	**0.755 (0.731, 0.782)**	**0.761 (0.735, 0.786)**	0.842 (0.812, 0.875)	**17.211 (12.106, 25.976)**	**0.802 (0.773, 0.831)**

**Table 3 bioengineering-10-00946-t003:** Maximum calibration errors of the two-class model (predicting “present” or “absent”) with equivocal images considered “absent”, two-class model (disregarding equivocal) with equivocal images disregarded from training and testing, and the three-class model (predicting “present”, “absent” and “equivocal”).

Experiment	Loss Function	MCE
Two-Class Model	Cross-Entropy Loss	0.424
	Focal Loss	0.385
Two-Class Model: Disregarding Equivocal	Cross-Entropy Loss	0.479
	Focal Loss	0.318
Three-Class Model	Cross-Entropy Loss	0.408
	Focal Loss	0.240
	Cross-Entropy Loss with probability targets	**0.150**

**Table 4 bioengineering-10-00946-t004:** Performance Metrics for the three-class model trained with cross-entropy loss using uncertainty-aware probability targets on the MIMIC-CXR external validation set. A 95% CI is reported and was calculated by bootstrap resampling 1000 times.

Experiment	AUROC	AUPRC	F-Score	Precision	Sensitivity	Specificity	Diagnostic Odds Ratio	Balanced Accuracy
Three-Class Model: Cross-Entropy Loss with Probability Targets	0.834 (0.811, 0.858)	0.898 (0.873, 0.917)	0.658 (0.627, 0.685)	0.729 (0.704, 0.749)	0.727 (0.703, 0.747)	0.955 (0.929, 0.973)	2.180 (1.801, 1.00)	0.841 (0.816, 0.860)

## Data Availability

Implementation code for this work is available at https://github.com/Kamaleswaran-Lab/CXaRds, accessed 1 August 2023. Data can be available by contacting the authors.

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
