# Peer review of "Uncertainty-Aware Convolutional Neural Network for Identifying Bilateral Opacities on Chest X-rays: A Tool to Aid Diagnosis of Acute Respiratory Distress Syndrome"

_bioengineering, 2023, doi:10.3390/bioengineering10080946_

Round 1

Reviewer 1 Report

The present study about convolutional neural networks for the identification of bilateral opacities on chest X-rays is very interesting and should be published.

However, the manuscript preparation seems to suffer under a very short presentation of the ML part. Especially for CNNs there is a huge amount of possibilities to adapt the network model to the current needs, i.e. for increasing the hit rate to the desired classification to be read out from chest X-ray images.

Is it possible to increase the described hit rate even more? It should be if we look in the appropriate literature. In the last years, a lot of studies for Covid identification were published about CNNs and X-ray images. It would be worth to use all this collected knowledge for the present study. The authors cited also three papers about Covid studies, however, the studies from which they could profit are not included.

E..g. the studies contain further developed variants of CNNs for gaining information out of chest X-rays:

Islam, M. Z., Islam, M. M. & Asraf, A. A combined deep CNN-LSTM network for the detection of novel coronavirus (Covid-19) using X-ray images. Inform. Med. Unlocked 20, 100412 (2020).

Saha, P., Sadi, M. S. & Islam, M. M. EMCNet: Automated COVID-19 diagnosis from X-ray images using convolutional neural network and ensemble of machine learning classifiers. Inform. Med. Unlocked 22, 100505 (2021).

Gulakala, R., Markert, B., Stoffel, M., Rapid diagnosis of Covid-19 infections by a progressively growing GAN and CNN optimisation. Computer Methods and Programs in Biomedicine 229, 107262, 2023,

Gulakala, R., Markert, B., Stoffel, M., Generative adversarial network based data augmentation for CNN based detection of Covid-19, Scientific Reports 12 (1), 19186, 2022

Heidari, M. et al. Improving the performance of CNN to predict the likelihood of Covid-19 using chest X-ray images with preprocessing algorithms. Int. J. Med. Inform. 144, 104284 (2020).

Lakhani, P. & Sundaram, B. Deep learning at chest radiography: Automated classification of pulmonary tuberculosis by using convolutional neural networks. Radiology 284, 574–582 (2017).

One of several new approaches is to implement generative learning for data augmentation. The authors should discuss, what is specially important for their study and why they decided to proceed in the way they did. However, they should describe that they are aware of the current state in literature and cite more articles like the mentioned ones. It could help to make use of this information to increase their accuracy.

The manuscript is recommended for publication after minor revisions.

Reviewer 2 Report

This study proposes a prediction model for ARDS based on the CNN model.

The manuscript is well presented and well described.

I have some suggestions regarding the study:

1. The performance evaluation metrics used in the study are AUROC, AUPRC, DOR, F-score, Precision, Sensitivity, and Specificity. There is no explanation about the formulas of F-score, Precision, Sensitivity, or Specificity in the manuscript. Please add the formula of F-score, Precision, Sensitivity, and Specificity.

2. Please add the accuracy measurement and explain its formula in the "2.3. Performance Metrics" subsection.

3. Have the authors applied k-fold cross-validation? I would like to recommend the authors apply 10-fold cross-validation. Please apply it and compare the results with the existing results.

4. Please explain the clinical relevance of the proposed study in the discussion section.

5. The performance results of the proposed model are less than 90% for all performance evaluation metrics. It would be great if the authors mentioned possible solutions to improve the performance of the proposed model.

Round 2

Reviewer 2 Report

All the concerns and suggestions have been addressed.